# COVID-19 and HIV: Clinical Outcomes among Hospitalized Patients in the United States

**DOI:** 10.3390/biomedicines11071904

**Published:** 2023-07-05

**Authors:** Zohaa Faiz, Mohammed A. Quazi, Neel Vahil, Charles M. Barrows, Hafiz Abdullah Ikram, Adeel Nasrullah, Asif Farooq, Karthik Gangu, Abu Baker Sheikh

**Affiliations:** 1Department of Medicine, School of Medicine, Aga Khan University, Karachi 74000, Pakistan; zohaa.faiz22@alumni.aku.edu; 2Department of Mathematics and Statistics, University of New Mexico, Albuquerque, NM 87106, USA; maquazi@salud.unm.edu; 3Department of Internal Medicine, University of New Mexico, Albuquerque, NM 87106, USA; nvahil@salud.unm.edu (N.V.); cmbarrows@salud.unm.edu (C.M.B.); haikram@unm.edu (H.A.I.); 4Division of Pulmonology and Critical Care, Allegheny Health Network, Pittsburg, PA 15212, USA; adeel.nasrullah@ahn.org; 5Department of Family and Community Medicine, Texas Tech Health Sciences Center, Lubbock, TX 79409, USA; afarooq76@gmail.com; 6Department of Internal Medicine, University of Kansas Medical Center, Kansas City, KS 66160, USA; kgangu2@kumc.edu

**Keywords:** COVID-19, HIV, mortality, prevalence, complications, United States, National Inpatient Sample

## Abstract

The concurrence of HIV and COVID-19 yields unique challenges and considerations for healthcare providers, patients living with HIV, and healthcare systems at-large. Persons living with HIV may face a higher risk of acquiring SARS-CoV-2 infection and experiencing worse clinical outcomes compared to those without. Notably, COVID-19 may have a disproportionate impact on historically disadvantaged populations, including African Americans and those stratified in a lower socio-economic status. Using the National Inpatient Sample (NIS) database, we compared patients with a diagnosis of both HIV and COVID-19 and those who exclusively had a diagnosis of COVID-19. The primary outcome was in-hospital mortality. Secondary outcomes were intubation rate and vasopressor use; acute MI, acute kidney injury (AKI); AKI requiring hemodialysis (HD); venous thromboembolism (VTE); septic shock and cardiac arrest; length of stay; financial burden on healthcare; and resource utilization. A total of 1,572,815 patients were included in this study; a COVID-19-positive sample that did not have HIV (*n* = 1,564,875, 99.4%) and another sample with HIV and COVID-19 (*n* = 7940, 0.56%). Patients with COVID-19 and HIV did not have a significant difference in mortality compared to COVID-19 alone (10.2% vs. 11.3%, respectively, *p* = 0.35); however, that patient cohort did have a significantly higher rate of AKI (33.6% vs. 28.6%, aOR: 1.26 [95% CI 1.13–1.41], *p* < 0.001). Given the complex interplay between HIV and COVID-19, more prospective studies investigating the factors such as the contribution of viral burden, CD4 cell count, and the details of patients’ anti-retroviral therapeutic regimens should be pursued.

## 1. Introduction

SARS-CoV-2, responsible for the COVID-19 pandemic, impacts various organ systems, with a global toll of 762,201,169 confirmed cases and 6 million deaths by April 2023 [1]. Vaccines have curbed incidence rates, hospitalizations, and mortality while reducing infection rates amidst emerging various variant strains [2,3]. Elderly patients and those with comorbidities, such as hypertension, diabetes, and immunosuppressed individuals, faced an increased risk of severe adverse effects from COVID-19 [4].

The intersection of HIV and COVID-19 presents a critical area of investigation due to the immunosuppressive characteristics of HIV, which impairs the host’s capacity to effectively combat infections [5]. The current literature on the relationship between HIV and COVID-19 is characterized by limited and discordant findings. Lee, K.W et al., and Karmen-Tuohy et al. presented contrasting evidence regarding COVID-19 severity in HIV patients, with the former suggesting less severe symptoms in advanced HIV stages and the latter indicating higher mortality rates in stage III HIV patients, underscoring the need for more rigorous research to elucidate the true impact of COVID-19 on this vulnerable population [6,7,8].

Researching HIV-positive communities has certain ethical ramifications as well because it may bring to light the violence, stigma, and difficulties these individuals endure daily. Additionally, if groups are vulnerable, it may prove an ethical challenge to give their free and informed consent to research [9].

With an estimated 38.4 million individuals living with HIV worldwide and 1.2 million people in the US alone, a thorough understanding of COVID-19’s consequences on their health is indispensable for informing clinical practice, public health strategies, and policy-making [10]. By conducting this study, we aim to provide robust evidence that could serve as a cornerstone for future research and interventions, ultimately enhancing the care and well-being of millions of people affected by both HIV and COVID-19.

This study aims to assess outcomes in COVID-19 patients with and without HIV using the National Inpatient Sample (NIS), focusing on in-hospital mortality as the primary outcome. Secondary outcomes encompass a range of clinical indicators, including mechanical ventilation, venous thromboembolism, acute liver failure, sudden cardiac arrest, cardiogenic shock, and acute kidney injury requiring hemodialysis, among others.

## 2. Materials and Methods

In this retrospective study, we made use of the NIS Healthcare Cost Utilization Project (HCUP) database, which is supported by the Agency for Healthcare Research and Quality (AHRQ). This database serves as a representative sample of discharges from community hospitals in the United States, approximating about 20% of the total [11]. Specifically, we analyzed the 2020 National Inpatient Sample (NIS) data, which cover hospitalizations that took place from 1 January 2020 to 31 December 2020. The data became publicly available in October 2022. By accessing this database, we obtained de-identified billing and diagnostic codes from participating hospitals. It is important to note that the NIS dataset is in compliance with federal regulations and guidelines, as it does not involve the direct involvement of “human subjects” [12]. Consequently, it is exempt from requiring approval from an institutional review board.

## 3. Inclusion and Exclusion Criteria

This study included all hospitalized patients aged 18 years and older who were diagnosed with COVID-19 infection. To identify the patient samples and comorbid conditions, we relied on ICD-10 clinical modification (CM) codes, while in-hospital procedures were identified using ICD-10 procedure codes. For a comprehensive overview of the codes used, please refer to Appendix A.

## 4. Covariates

The NIS dataset provided valuable information on various aspects, including in-hospital outcomes, procedures, and discharge-related data. The variables in this study were categorized into three groups: patient level, hospital level, and illness severity. Patient-level variables encompassed age, race, gender, comorbidities, insurance status, income based on the patient’s zip code, and disposition. Hospital-level variables included location, teaching status, bed size, and region. Illness severity variables consisted of the length of hospital stay (LOS), mortality rate, hospitalization cost, Elixhauser comorbidity score, in-hospital complications like mechanical ventilation, mechanical circulatory support, vasopressor use, cardiac arrest, cardiogenic shock, acute venous thromboembolism (VTE), acute kidney injury, acute kidney injury requiring hemodialysis, and acute liver failure.

## 5. Study Outcomes

The primary outcome of interest in this study was in-hospital mortality. Secondary outcomes included (a) invasive and non-invasive mechanical ventilation, (b) vasopressor use, acute kidney injury (AKI), AKI requiring hemodialysis (HD), and VTE, (c) presence of cardiogenic shock and cardiac arrest, (d) VTE, acute liver failure, and cerebrovascular accident, (e) length of hospital stay, (f) financial impact on the healthcare system, and utilization of resources.

## 6. Statistical Methods

The Python programming language was utilized for data analysis and statistical modeling. Data curation required the use of SAS.

The initial sample consisted of 6.47 million unweighted observations, and after applying weights, the sample size increased to approximately 32.3 million discharges for the year 2020. Among these, there were 1.67 million patients who were admitted with COVID-19. However, due to missing values for the variables of interest, the sample size for this study was restricted to 1,572,815 records. Out of the COVID-19-positive samples, 1,564,875 individuals did not have HIV, while 7940 individuals had HIV.

The chi-square test for independence was used to test relationships between the two cohorts and the categorical variables reported in Table 1. Small *p*-values imply that the two variables are not independent of each other. For example, gender and cohort variables are not independent, and there exists a relationship between them (*p*-value < 0.001). Simple linear regression was used to identify independent variables (*p* ≤ 0.2) for continuous responses (e.g., length of stay and total charges) to build a multivariate regression model, and logistic regression was used to identify independent variables (*p* ≤ 0.2) for binary response variables (e.g., mechanical circulatory support, AKI, etc.). A secondary analysis on the propensity-matched sample was run to corroborate the findings from the statistical models on the unmatched sample because our case group, which is the COVID-19 and HIV, had a much larger sample than the control group, which is the COVID-19 and HIV+. 

As covariates, baseline demographics and Elixhauser comorbidities were used for propensity matching in R. The regression models were built again for the matched cohorts as explained for the unmatched cohorts.

## 7. Results

### 7.1. Demographics and Baseline Comorbidities

In our study population, a total of 1,572,815 hospitalized COVID-19 patients, out of which 7940 (0.50%) had a concomitant diagnosis of HIV, were included. COVID-19 patients with HIV were more often males (68.1% vs. 51.7%, *p* < 0.001) and had a greater proportion of African Americans (55.9% vs. 18.9%, *p* < 0.001). HIV with COVID-19 infection was significantly more prevalent among patients aged 50–69 years (55.2% vs. 37.0%, *p* < 0.001), 30–49 years (28.1% vs. 16.7%, *p* < 0.001), and 18–29 years (5.2% vs. 4.9%, *p* < 0.001), with mean ages of 54.3 ± 13.5 years for females and 54.4 ± 13.6 years for males. This cohort was also more likely to have a median household income below 50,000 USD (48.1% vs. 34.0%, *p* < 0.001) compared to COVID-19 patients without HIV.

Patients with COVID-19 with HIV were more likely to smoke (33.8% vs. 25.5%, *p* < 0.001), consume alcohol (3.5% vs. 2.3%, *p* < 0.001), and have a history of drug abuse (7.7% vs. 1.9% *p* < 0.001). They were also more likely to have chronic pulmonary disease (24.4% vs. 21.6% *p* = 0.006) and suffer from depression (13.4% vs. 10.9% *p* = 0.001). However, patients with COVID-19 alone had significantly higher rates of obesity (26.5% vs. 21.6%, *p* < 0.001), hypertension (64.9% vs. 60.5%, *p* < 0.001), diabetes (40.0% vs. 37.5%, *p* = 0.033) and coronary artery disease (CAD) (17.9 vs. 12.4%, *p* < 0.001). This cohort of patients was also more likely to suffer from peripheral vascular disease (4.0% vs. 2.8%, *p* = 0.009), hypothyroidism (13.3% vs. 5.6%, *p* < 0.001), autoimmune diseases (3.1% vs. 1.6%, *p* < 0.001), and dementia (12.1% vs. 6.2%, *p* < 0.001). There was no significant difference in regard to chronic kidney disease (CKD) and myocardial infarction (MI) between the two cohorts (Table 1).

There were some variations in the geographical spread of the patients. The majority of patients in both groups were treated at urban teaching hospitals (83.0% vs. 71.5%, respectively), and a larger proportion were recipients of Medicare (37.9% vs. 50.1%, respectively, *p* < 0.001). The baseline demographics of the study cohorts are summarized in Table 1. The initial baseline characteristics of the matched cohort, post propensity matching, are detailed in Appendix A.

### 7.2. In-Hospital Mortality

We conducted a multivariate logistic regression, adjusting for several elements like age, hospital bed capacity, race, gender, hospital location, hospital teaching status, hospital’s regional placement, median household income, anticipated primary payer, and Elixhauser comorbidities (as seen in Table 2). We found no significant disparity in in-hospital mortality between COVID-19 patients not living with HIV and those having both COVID-19 and HIV (10.2%, adjusted OR [aOR]: 1.1 [95% CI 0.9–1.2], *p* = 0.55) (Table 2).

We also implemented propensity matching considering variables such as patient age, gender, race, income, insurance status, and Elixhauser (refer to Appendix A). After the Propensity Score Matching (PSM) process, each group (those with HIV and those without) comprised 7940 patients (Appendix A). The total number of patients with in-hospital mortality was 1710, with 900 (11.3%) from the COVID and HIV− group and 810 (10.2%) from the COVID and HIV+ group. The aOR was calculated to be 0.88 (95% CI 0.69–1.13), with a *p*-value of 0.35, indicating that the difference in mortality between the two groups was not statistically significant (Table 3).

### 7.3. In-Hospital Complications

The incidence of acute kidney injury (AKI) was significantly higher in patients with COVID-19 and HIV compared to those with COVID-19 alone (33.6% vs. 28.6%, aOR: 1.26 [95% CI 1.13–1.41], *p* < 0.001). Similarly, hemodialysis was more common in patients with COVID-19 and HIV compared to those with COVID-19 alone (9.0% vs. 5.1%, aOR: 1.24 [95% CI 1.04–1.49], *p* = 0.015). Interestingly, the incidence of cerebrovascular accident (CVA) was significantly lower in patients with COVID-19 and HIV compared to those with COVID-19 alone (1.0% vs. 1.7%, aOR: 0.56 [95% CI 0.34–0.93], *p* = 0.025) (Table 2).

The study also analyzed the incidence of other complications, including acute liver failure (1.2% vs. 1.0%, aOR: 1.05 [95% CI 0.67–1.64], *p* = 0.807), sudden cardiac arrest (2.4% vs. 2.7%, aOR: 0.82 [95% CI 0.59–1.14], *p* = 0.256), vasopressor use (2.83% vs. 2.6%, aOR: 1.02 [95% CI 0.75–1.38], *p* = 0.879), mechanical circulatory support (0.12% vs. 0.3%, aOR: 0.32 [95% CI 0.08–1.31], *p* = 0.116), venous thromboembolism (VTE) (5.2% vs. 4.6%, aOR: 1.00 [95% CI 0.80–1.25], *p* = 0.965), cardiogenic shock (0.7% vs. 0.6%, aOR: 0.95 [95% CI 0.52–1.74], *p* = 0.889), invasive (10.9% vs. 11.0%, aOR: 0.91 [95% CI 0.78–1.07], *p* = 0.290), and non-invasive mechanical ventilation (4.1% vs. 5.6%, aOR: 0.85 [95% CI 0.66–1.09], *p* = 0.207). However, there were no significant differences in the incidence of these complications between the two groups of patients (Table 2).

After PSM, patients with HIV and COVID-19 continued to have a greater incidence of AKI (33.62% vs. 29.8%, aOR: 1.24 [95% CI 1.06–1.46], *p* = 0.005). However, again there was no significant difference in usage of invasive mechanical ventilation (0.1% vs. 0.4%, aOR: 0.32 [95% CI 0.06–1.62], *p* = 0.172), hemodialysis (9.0% vs. 9.3%, aOR: 0.98 [95% CI 0.76–1.26], *p* = 0.888), non-invasive mechanical ventilation (4.1% vs. 5.6%, aOR: 0.8 [95% CI 0.55–1.06], *p* = 0.115), vasopressor use (2.8% vs. 3.4%, aOR: 0.84 [95% CI 0.56–1.25], *p* = 0.399), and mechanical circulatory support (0.1% vs. 0.4%, aOR: 0.32 [95% CI 0.06–1.62], *p* = 0.172). Furthermore, there was also no significant difference in rates of venous thromboembolism (VTE) (5.2% vs. 5.7%, aOR: 0.80 (95% CI 0.58–1.12), *p* = 0.214), cardiogenic shock (0.7% vs. 1.2%, aOR: 0.51 [95% CI 0.23–1.13], *p* = 0.099), cardiac arrest (2.4% vs. 2.8%, aOR: 0.73 [95% CI 0.45–1.18], *p* = 0.200), and cerebrovascular accident (CVA) (1.0% vs. 1.8%, aOR: 0.55 [95% CI 0.29–1.02], *p* = 0.058) (Table 3).

**Table 3 biomedicines-11-01904-t003:** In-hospital outcomes of propensity-matched sample.

Variable	COVID-19 Patients without HIV	COVID-19 Patients with HIV	*p* Value
In hospital mortality (*n* = 1710)	11.33%	10.20%	0.349
	Adjusted odds ratio	0.88 (95% CI 0.69–1.13)	
Vasopressor use	3.40%	2.83%	0.399
	Adjusted odds ratio	0.84 (95% CI 0.56–1.25)	
Non-Invasive Mechanical Ventilation	5.54%	4.15%	0.115
	Adjusted odds ratio	0.76 (95% CI 0.55–1.06)	
Invasive Mechanical Ventilation	13.22%	10.89%	0.051
	Adjusted odds ratio	0.79 (95% CI 0.62–1.00)	
Sudden Cardiac Arrest	2.77%	2.39%	0.2
	Adjusted odds ratio	0.73 (95% CI 0.45–1.18)	
Cardiogenic shock	1.19%	0.69%	0.099
	Adjusted odds ratio	0.51 (95% CI 0.23–1.13)	
Mechanical Circulatory Support	0.37%	0.12%	0.172
	Adjusted odds ratio	0.32 (95% CI 0.06–1.62)	
Venous Thromboembolism	5.66%	5.16%	0.214
	Adjusted odds ratio	0.80 (95% CI 0.58–1.12)	
Cerebrovascular Accident	1.82%	1.00%	0.058
	Adjusted odds ratio	0.55 (95% CI 0.29–1.02)	
Acute Kidney Injury	29.84%	33.62%	0.005
	Adjusted odds ratio	1.24 (95% CI 1.06–1.46)	
Hemodialysis	9.31%	9.00%	0.888
	Adjusted odds ratio	0.98 (95% CI 0.76–1.26)	
Acute Liver Failure	1.38%	1.25%	0.449
	Adjusted odds ratio	0.77 (95% CI 0.40–1.49)	
Mean total hospitalization charges ($)	95,475	102,309	0.853
	Adjusted total charge	1224.10 higher for HIV+	
Mean length of stay (days)	8.51	8.67	0.463
	Adjusted length of stay	0.28 days lower for HIV+	

### 7.4. Predictors of Mortality

A forest plot (Appendix A) was created to map predictors of mortality in HIV-1 patients. The plot determined that acute liver failure HR: 4.5 ([95% CI 2.4–8.5], *p* < 0.001), sudden cardiac arrest HR: 4.2 ([95% CI 2.6–6.7], *p* < 0.001), invasive mechanical ventilation HR: 2.1 ([95% CI 1.4–3.2], *p* < 0.001), and noninvasive mechanical ventilation HR: 2.5 ([95% CI 1.6–4.1], *p* < 0.001) were significant predictors of mortality in HIV-1 patients with COVID-19 infection.

### 7.5. In-Hospital Quality Measures and Disposition

Patients with COVID-19 and HIV did not have a statistically significant increased mean length of stay (8.67 days vs. 8.02 days, adjusted length of stay 0.28 days higher for those with HIV, *p* = 0.214) compared to COVID-19 patients without HIV infection. This cohort of patients also did not have a statistically significant difference in total hospitalization charge (USD 102,309 vs. USD 91,696, adjusted total charge USD 5603 higher, *p* = 0.193) Table 2.

Patients with both COVID-19 and HIV had higher percentages of discharge against medical advice (3.0% vs. 1.1%, *p* < 0.001) and higher percentages of discharge to routine care (57.8% vs. 50.9%, *p* < 0.001). They also experienced lower percentages of hospital deaths (10.1% vs. 13.4% *p* < 0.001), transfers to other healthcare facilities (15.6 vs. 18.6 *p* < 0.001), discharges to home health care (11.3% vs. 13.0%, *p* < 0.001), and admissions to short-term hospitals (2.0% vs. 3.0%, *p* < 0.001).

After PSM, HIV patients who were COVID-19-positive continued to have an increased mean length of stay (8.7 days vs. 8.5 days, adjusted length of stay 0.28 days lower, *p* = 0.463) and a higher mean total hospitalization charge (USD 102,309 vs. USD 95,475, adjusted total charge USD 1224 higher, *p* = 0.853) than COVID-19-positive HIV-negative patients (Table 3). However, both differences were statistically insignificant.

Similarly, after PSM, patients with both COVID-19 and HIV had higher percentages of discharge against medical advice (3.0% vs. 2.0%, *p* = 0.178), lower percentages of death in the hospital (10.2% vs. 11.3%, *p* = 0.178), higher percentages of transfer to other healthcare facilities (15.6% vs. 13.5%, *p* = 0.178), similar percentages of discharge to home health care (11.3% vs. 11.9%, *p* = 0.178), lower percentages of discharge to routine care (57.8% vs. 58.8%, *p* = 0.178), and lower percentages of transfer to a short-term hospital (1.9% vs. 2.5%, *p* = 0.178).

## 8. Discussion

Major findings of our study include: (1) There was no significant increase in in-hospital mortality among COVID-19 patients who are also HIV positive when compared to their HIV-negative counterparts. (2) COVID-19 patients with HIV experienced significantly higher rates of AKI-necessitating hemodialysis (included or not). (3) There was a racial disparity in the composition of patients who had both COVID-19 and HIV. This group was more likely to be African Americans with a lower income bracket.

In this retrospective analysis, we identified 1,572,815 hospitalized patients diagnosed with COVID-19 between 1 January 2020 and 31 December 2020, out of which 7940 (0.5%) with a concomitant diagnosis of HIV and COVID-19 infection. According to thirteen independent research conducted in the United States, the reported pooled prevalence of COVID-19 patients with HIV is 12.9% (CI 7.7–19.5) [13]. We also included 1,564,875 COVID-19 patients without HIV during the same duration.

The results of the study revealed that patients with HIV and COVID-19 were more likely to be male and African American, in accordance with a systematic review completed by Mirzaei et al. [14]. Because of existing health inequities, minorities are at a higher risk of developing chronic illnesses [15]. This can be accounted for by various factors, which include lack of care due to significant disparities among African Americans when compared to their white and more socioeconomically-abled counterparts [16]. Another reason has been found to be medical distrust among Black and Latino individuals secondary to discriminatory events that continue to be reported in different parts of the country [17]. Various studies show that males are predisposed to serious complications from COVID-19 to a greater extent [18,19]. This finding has been attributed to the role of testosterone as a pro-inflammatory hormone, as opposed to estrogen, which has more anti-inflammatory effects [20]. These differences could be due to androgens on the expression of ACE2 and TMPRSS2, which act as entry points for the SARS-CoV-2 virus into host cells [21,22].

HIV patients with COVID-19 also tended to be of a younger age group with lower median household incomes compared to COVID-19 patients without HIV. A younger age group was also found significant in a systematic review conducted by Ssentongo et al. Their findings revealed that the average age of HIV patients with COVID-19 was 10 years younger than COVID-19 patients who did not have HIV (55 years vs. 65 years) [23]. This substantial age difference may be influenced by a higher prevalence of behaviors in younger people that increase the risk of HIV transmission. These behaviors can include unsafe sexual practices and sharing of injection equipment, often associated with substance misuse, including smoking, alcohol, and recreational drug misuse [24]. All these study variables were also significantly greater in our HIV & COVID-19 patient group, thus strengthening our findings.

The presence of chronic pulmonary disease, found in a greater proportion of patients who are immunocompromised [25], was also strongly associated in our HIV and COVID-19 cohort.

Interestingly, patients with COVID-19 alone had higher rates of obesity, hypertension, diabetes, CAD, peripheral vascular disease, hypothyroidism, autoimmune diseases, and dementia. These findings may be attributed to the group’s higher mean age and the presence of these comorbidities in an older population, especially those admitted with COVID-19 [4,26].

While our key objective was to investigate a discrepancy in in-hospital mortality between COVID-19 patients with and without HIV, we did not identify a considerable difference between these two groups. Some of the literature has suggested a greater mortality rate among patients with co-infection. Mellor et al.’s review of 19 studies discovered a significantly higher risk of COVID-19 mortality in PLWH (hazard ratio 1.95, 95% CI: 1.62–2.34) [27]. Additionally, a meta-analysis by Ssentongo and colleagues also reported a higher risk of death in the co-infection group (RR 1.78, 95% CI 1.21–2.60) [23]. However, another meta-analysis completed by Dzinamarira et al. in 2022 reported no significant association between these two groups (RR 1.07, 95% CI 0.86–1.32) [28]. It should be noted here that these data are only the result of an epidemiological evaluation. These findings have several implications. The results can be confounded by the link between older age, multimorbidity, and concomitant COVID-19 severe enough to cause hospital admission of the patient. Patients with COVID-19 alone are at substantially increased risk of mortality based on the presence of other comorbidities [14], which were reported in a higher frequency in this group, as discussed above. According to existing clinical evidence, the leading mortality risk factors not specific to HIV are advanced age and prevalence of chronic diseases, including heart diseases, diabetes, chronic pulmonary disease, and hypertension [29]. Some studies have revealed a therapeutic effect of anti-retroviral therapy (ART) on the severity of COVID-19 infection [30,31], which could be another possible confounding factor leading to this result.

Another probable explanation is that HIV infection creates a state of immunosuppression that prevents the host from triggering the strong and robust immune response that drives a cytokine storm. The cytokine storm generated is primarily responsible for the severity of the clinical manifestations of COVID-19 infection. So, patients with HIV infection would be more likely to have less severe outcomes with concomitant COVID-19 infection [31,32].

Patients with COVID-19 plus HIV had a higher occurrence of acute kidney injury and hemodialysis than those with COVID-19 alone. While COVID-19 has been known to cause direct kidney damage secondary to an inflammatory response it instigates [33], HIV has been known to affect the renal system through several mechanisms. These include chronic inflammation, immunological dysfunction, and adverse effects of anti-retroviral drugs [34,35]. Hence, the combined effects of these infections may have been a pivot in the greater prevalence of AKI in the first case group, as supported by Fisher et al. [33]. Both HIV and COVID-19 have been reported to cause kidney damage, and the combined effect might lead to more severe outcomes.

Importantly, results to the contrary have been demonstrated in the literature [33]. Fisher et al. showed that patients with HIV and COVID-19 did not have different mortality outcomes than those with COVID-19 alone. However, the authors note that although this finding is unexpected, the lack of an association between this cohort and increased AKIs may be related to the likelihood of comorbidities such as diabetes, hypertension, and chronic kidney disease (CKD) being present in patients who ultimately develop in-hospital kidney injuries. Another possibility, which is also present in our study, is that due to the small size of the HIV and COVID-19, subgroups were underpowered, making it challenging to detect statistically significant differences between cohorts. 

Similarly, peculiar results were reflected in a study by Patel et al., 2021, who showed that rates of intubation were higher in patients with higher CD4 counts [36]. This suggests that a more robust immune response in patients with HIV may actually be deleterious compared to those who are more immunocompromised, particularly when augmented by the immune hyperactivation characteristic of COVID-19. It is plausible that a similar mechanism may explain the lower rates of AKI found by Fisher et al. 

Acute liver failure is a well-known complication of HIV infection, and it is associated with increased mortality rates in HIV patients, especially those with advanced illness or co-infection with other viruses, including COVID-19, hepatitis B or C. Liver disease is one of the primary causes of non-AIDS-related death, accounting for 13–18% of all-cause mortality among HIV-infected individuals [37,38]. Furthermore, acute liver failure has been associated with anti-retroviral drugs with known hepatotoxicity [39]. Sudden cardiac arrest is another serious complication that can occur in HIV-infected individuals. They are more likely to suffer from myocardial fibrosis secondary to immune activation and chronic inflammation [40]. Hence, patients diagnosed with HIV have been found to be two times more likely than the rest of the population to perish from sudden cardiac arrest [41]. In terms of invasive and non-invasive mechanical ventilation (MV) and its impact on the life expectancy of HIV patients, Karmen-Tuohy et al. reported a trend toward an increased need for ICU admission and ventilation as well as longer stays in hospital compared to matched HIV-negative patients [8]. Pathak et al. found a substantial increase in mortality of HIV patients after MV (44%) compared to individuals without HIV (22%; *p* = 0.01) [42]. Studies have also linked the increased incidence of ventilator-associated pneumonia (VAP), a significant factor of mortality in MV patients, with the presence of HIV-AIDS. This relationship can be explained, in part, by the immunosuppression caused by the disease, as well as the longer duration of MV in such patients [42,43].

The predictors of mortality—acute liver failure, sudden cardiac arrest, and both invasive and non-invasive mechanical ventilation—suggest that severe systemic illness and multi-organ involvement increase the risk of death in COVID-19 patients with HIV. It is important to note that these predictors are often indicators of severe COVID-19 disease. From the immune dysregulation to the chronic inflammation and hepatotoxic agents administered, HIV dramatically increases the risk of organ dysfunction [44]. COVID-19, particularly in its severe form, can cause extensive multiple organ failure owing to both a direct invasion of the virus into host viral tissue as well as a cytokine storm that yields sustained inflammation [45]. Identifying persons with HIV who are at risk for organ dysfunction due to drug toxicity, immune dysregulation, or chronic inflammatory states can help providers identify subgroups of patients who may be at higher risk of multiple organ dysfunction to prompt early recognition and more tailored management of severe COVID-19 symptoms in this population to potentially reduce the mortality risk. 

We acknowledge that our study’s database limitations preclude the stratification of patients based on CD-4 cell counts. Such an analysis could have nuanced our findings and allowed for an assessment of the effects of COVID-19 on HIV patients with optimally managed conditions. As underscored in research by Rossenheim et al., even well-controlled HIV infections often present an elevated level of complement activation, indicative of persistent hyperinflammation [46]. While anti-retroviral therapy (ART) can manage this to some extent, it does not fully resolve the chronic inflammatory state. Such unmitigated inflammation has been linked to a 2–4 fold higher mortality risk from non-AIDS-defining events, including cardiovascular disease, and is also associated with metabolic disorders, bone disease, kidney disease, and neurocognitive dysfunction [46].

## 9. Limitations

Our study, while offering key insights into the health outcomes of co-infected HIV and COVID-19 patients, is subject to several limitations. Firstly, the reliance on the NIS database, although a comprehensive resource, carries inherent drawbacks related to its retrospective design. Potential coding inaccuracies, lack of detailed data, and the inability to track patient data over time are among these challenges. Secondly, our study design allows for correlation but not causation, and despite our efforts to minimize confounding through propensity score matching, the possibility of residual confounding persists. Thirdly, the validity of HIV and COVID-19 diagnoses based exclusively on ICD codes may be called into question. Fourthly, the NIS database lacks certain crucial clinical data, such as HIV disease duration, viral load, CD4 count, and specifics of anti-retroviral therapy. Additionally, our study does not consider adherence to HIV treatment or the severity of immunosuppression which may have a significant influence on patient outcomes. Finally, our findings may not apply to patients outside of the hospital setting or to healthcare systems outside the United States. Despite these constraints, our research contributes valuable information to the emerging field of HIV and COVID-19 interplay.

## 10. Future Directions

Our investigation, with its focus on clinical consequences in patients co-diagnosed with HIV and COVID-19, delineates pivotal avenues for prospective scholarly pursuits. Subsequent investigations should consider adopting a prospective methodology, thereby facilitating a more exhaustive accumulation of patient data, encompassing treatment compliance, intricate clinical variables such as HIV disease tenure, viral burden, CD4 cell count, and specifics pertaining to anti-retroviral therapeutic regimes. Such granularity of data can yield profound insights into the interdependent dynamics of these diseases and their bearing on patient prognosis. The discerned racial and socio-economic bifurcation within the demographic diagnosed with both HIV and COVID-19 warrants a deeper inquisition into the causative factors underpinning this inequality, potentially guiding the formulation of more egalitarian healthcare policies and strategies. Future research endeavors should transcend the confines of the inpatient environment to include outpatient cohorts, thereby enhancing the applicability of these findings to a broader spectrum.

Our study augments the extant body of literature by delivering an extensive, nationally representative analysis of clinical outcomes in HIV-positive patients afflicted with COVID-19. Notwithstanding the innate limitations, our exploration has successfully bridged an information chasm, laying the groundwork for subsequent rigorous investigations to unravel the complexities associated with managing patients concurrently diagnosed with HIV and COVID-19. The conclusions drawn from this study bear the significant potential to inform clinical praxis and shape public health stratagems. Highlighting the importance of renal management and the need to address health inequities within this susceptible population.

## 11. Conclusions

While our study did not find a higher rate of in-hospital mortality associated with co-infection of COVID-19 and HIV, it did reveal a markedly higher rate of AKI in the co-infected cohort compared to those with COVID-19 alone. Significantly, our study identified substantial racial and socio-economic disparities within the co-infected patient population, with a majority of patients being African American and belonging to lower socio-economic strata. This finding highlights the critical interplay between socio-economic factors and health outcomes and underscores the need for multi-dimensional strategies that address these concurrent public health crises. These strategies should include not only medical interventions but also socio-economic measures aimed at mitigating the effects of these diseases on the most vulnerable populations. It is crucial to acknowledge that our conclusions are based primarily on an epidemiological evaluation and do not consider certain key viro-immunological factors due to the lack of specific patient data. The unavailability of detailed information such as viral load, CD4 count, and specific anti-retroviral therapy regimen limits our ability to draw definitive conclusions regarding the clinical interaction between HIV and COVID-19. Given these limitations, our findings should be viewed as an initial broad epidemiological investigation of the outcomes of individuals co-infected with HIV and COVID-19. Our study suggests areas that warrant more in-depth examination and points towards the necessity for future research to further elucidate the relationship between these two infections. This includes identifying specific HIV-positive subgroups that may be at higher risk of adverse clinical outcomes from COVID-19, thus contributing to a more comprehensive understanding of these intersecting diseases.

## Figures and Tables

**Table 1 biomedicines-11-01904-t001:** Patient-level characteristics for COVID-19-positive patients with HIV and COVID-19-positive patients without HIV.

Characteristics	COVID and HIV−	COVID and HIV+	*p* Value
*n* = 1,572,815	*n* = 1,564,875 (99.49%)	*n* = 7940 (0.50%)	
Gender (%)			<0.001
Female	755,315 (48.27%)	2535 (31.93%)	
Male	809,560 (51.73%)	5405 (68.07%)	
Mean Age Years (SD)			
Female	63.1 (18.87)	54.3 (13.5)	
Male	63.6 (16.3)	54.3 (13.55)	
AGE Groups (%)			<0.001
18–29	77,330 (4.94%)	410 (5.16%)	
30–49	261,230 (16.69%)	2230 (28.09%)	
50–69	578,955 (37.00%)	4385 (55.23%)	
≥70	647,360 (41.37%)	915 (11.52%)	
RACE (%)			<0.001
Asian or Pacific	51,310 (3.28%)	105 (1.32%)	
Black	295,375 (18.88%)	4435 (55.86%)	
Hispanic	336,180 (21.48%)	1495 (18.83%)	
Native American	14,795 (0.95%)	35 (0.44%)	
Other	67,025 (4.28%)	350 (4.41%)	
White	800,190 (51.13%)	1520 (19.14%)	
Median household income (%)			<0.001
≤49,999	532,735 (34.04%)	3820 (48.11%)	
50 k–64,999	424,230 (27.11%)	2015 (25.38%)	
65 k–85,999	347,705 (22.22%)	1255 (15.81%)	
≥86 k	260,205 (16.63%)	850 (10.71%)	
Insurance status (%)			<0.001
Medicaid	223,165 (14.26%)	2080 (26.20%)	
Medicare	797,635 (50.97%)	3010 (37.91%)	
No charge	4210 (0.27%)	35 (0.44%)	
Other	68,145 (4.35%)	405 (5.10%)	
Private Insurance	413,185 (26.40%)	1965 (24.75%)	
Self-pay	58,535 (3.74%)	445 (5.60%)	
Hospital division (%)			<0.001
East North Central	245,620 (15.70%)	990 (12.47%)	
East South Central	107,045 (6.84%)	510 (6.42%)	
Middle Atlantic	234,590 (14.99%)	850 (10.71%)	
Mountain	99,405 (6.35%)	360 (4.53%)	
New England	60,015 (3.84%)	480 (6.05%)	
Pacific	177,825 (11.36%)	420 (5.29%)	
South Atlantic	319,690 (20.43%)	3045 (38.35%)	
West North Central	96,945 (6.20%)	130 (1.64%)	
West South Central	223,740 (14.30%)	1155 (14.55%)	
Hospital bed size (%)			0.014
Large	730,350 (46.67%)	3955 (49.81%)	
Medium	454,085 (29.02%)	2275 (28.65%)	
Small	380,440 (24.31%)	1710 (21.54%)	
Hospital teaching status (%)			<0.001
Rural	150,015 (9.59%)	250 (3.15%)	
Urban nonteaching	296,205 (18.93%)	1100 (13.85%)	
Urban teaching	1,118,655 (71.49%)	6590 (83.00%)	
Comorbidities (%)			
CAD	279,830 (17.88%)	985 (12.40%)	<0.001
MI	65,770 (4.20%)	275 (3.46%)	0.142
HTN	1,015,640 (64.90%)	4805 (60.51%)	<0.001
Diabetes (2)	627,355 (40.08%)	2975 (37.46%)	0.033
Cancer (5)	65,375 (4.17%)	470 (5.91%)	<0.001
Obesity	414,250 (26.47%)	1715 (21.59%)	<0.001
Drug Abuse	28,975 (1.85%)	610 (7.68%)	<0.001
Smoking	398,885 (25.48%)	2685 (33.81%)	<0.001
Alcohol	35,420 (2.26%)	280 (3.52%)	<0.001
Chronic Pulmonary Disease	338,180 (21.61%)	1940 (24.43%)	0.006
Peripheral Vascular Disease	63,500 (4.05%)	220 (2.77%)	0.009
CKD	197,770 (12.63%)	1045 (13.16%)	0.531
Hypothyroidism	207,645 (13.26%)	445 (5.60%)	<0.001
Autoimmune	48,655 (3.10%)	125 (1.57%)	<0.001
Depression	170,615 (10.90%)	1065 (13.41%)	0.001
Dementia	189,265 (12.09%)	490 (6.17%)	<0.001

CAD: Coronary Artery Disease; MI: Myocardial Infarction; HTN: Hypertension; CKD: Chronic Kidney Disease.

**Table 2 biomedicines-11-01904-t002:** In-hospital outcomes for COVID-19-positive patients with HIV and COVID-19 patients without HIV.

Characteristics	COVID and HIV−	COVID and HIV+	*p* Value
*n* = 1,572,815	1,564,875 (99.5%)	7940 (0.5%)	
Disposition (%)			<0.001
Against medical advice	17,325 (1.11%)	240 (3.02%)	
Died in hospital	209,830 (13.41%)	810 (10.20%)	
Discharged alive unknown destination	900 (0.06%)	5 (0.06%)	
Home health care	203,380 (13.00%)	900 (11.34%)	
Routine	796,050 (50.87%)	4590 (57.81%)	
Transfer other	290,625 (18.57%)	1240 (15.62%)	
Transfer to short-term hospital	46,765 (2.99%)	155 (1.95%)	
COMORBIDITIES (%)			
Acute Liver Failure	16,125 (1.03%)	100 (1.25%)	0.807
	Adjusted odds ratio * = 1.05 (95% CI 0.67–1.64)		
Sudden Cardiac Arrest	41,560 (2.65%)	190	0.256
	Adjusted odds ratio * = 0.82 (95% CI 0.59–1.14)		
Mean total hospitalization charge ($)	$91,696.93	$102,309.39	0.193
	Adjusted total charge * = $5603.80 higher for HIV+		
Mean length of stay (days)	8.02	8.67	0.214
	Adjusted length of stay * = 0.28 days higher for HIV+		
In hospital mortality (*n* = 210,640)	209,830 (13.40%)	810 (10.2%)	0.547
	Adjusted odds ratio * = 1.05 (95% CI 0.89–1.24)		
Vasopressor use	40,965 (2.61%)	225 (2.83%)	0.879
	Adjusted odds ratio * = 1.02 (95% CI 0.75–1.38)		
Mechanical Circulatory Support	4220 (0.26%)	10 (0.12%)	0.116
	Adjusted odds ratio * = 0.32 (95% CI 0.08–1.31)		
AKI	447,470 (28.6%)	2670 (33.6%)	<0.001
	Adjusted odds ratio * = 1.26 (95% CI 1.13–1.41)		
VTE	72,680 (4.6%)	410 (5.16%)	0.965
	Adjusted odds ratio * = 1.00 (95% CI 0.80–1.25)		
Cardiogenic shock	9560 (0.61%)	55 (0.69%)	0.889
	Adjusted odds ratio * = 0.95 (95% CI 0.52–1.74)		
Hemodialysis	79,315 (5.1%)	715 (9%)	0.015
	Adjusted odds ratio * = 1.24 (95% CI 1.04–1.49)		
Invasive Mechanical Ventilation	171,645 (11.0%)	865 (10.9%)	0.290
	Adjusted odds ratio * = 0.91 (95% CI 0.78–1.07)		
Non-Invasive Mechanical Ventilation	88,095 (5.6%)	330 (4.1%)	0.207
	Adjusted odds ratio * = 0.85 (95% CI 0.66–1.09)		
CVA	26,000 (1.7%)	80 (1.0%)	0.025
	Adjusted odds ratio * = 0.56 (95% CI 0.34–0.93)		

AKI: Acute Kidney Injury, VTE: Venous Thromboembolism, CVA: Cerebrovascular Accident: * Adjusted for age, hospital bed size, race, gender, hospital location, hospital teaching status, hospital region, median household income, expected primary payer (insurance status), Elixhauser comorbidities.

## Data Availability

The data used in this study are from the National (Nationwide) Inpatient Sample for the year 2020, obtained from the Healthcare Cost and Utilization Project. The NIS is a publicly available database, and researchers interested in accessing the data can obtain it directly from the HCUP website (https://www.hcup-us.ahrq.gov/).

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
