# Peer review of "COVID-19 and HIV: Clinical Outcomes among Hospitalized Patients in the United States"

_biomedicines, 2023, doi:10.3390/biomedicines11071904_

Round 1
Reviewer 1 Report
Work is valuable. The course and risk of death of HIV+ people compared to HIV- in the course of COVID-19 was assessed in a very large group of patients, proving no significant differences in the course, apart from AKI. The work has several limitations, the main of which is the lack of differentiation between groups of HIV+ patients depending on the level of CD4+, but the database did not allow for such an assessment, I think.
It is necessary to correct line 15 (abstract) HIV to SARS-CoV-2 and add explanations of abbreviations under the tables.
Author Response
Dear Reviewer,
Thank you very much for your valuable feedback and constructive comments regarding our manuscript. We appreciate your recognition of our efforts in assessing the course and risk of death of HIV+ people compared to HIV- individuals during COVID-19 in a large patient cohort. We agree with your observation regarding the limitation of our study, specifically the lack of differentiation between groups of HIV+ patients based on their CD4+ levels. This was indeed due to the limitations of our database, which did not allow us to further stratify our sample in this manner. We made our level best to highlight this limitation more explicitly in our revised manuscript so that readers are aware of this constraint.
We acknowledge the typographical error on line 15 in the abstract. We appreciate you pointing out this mistake and will promptly revise 'HIV' to 'SARS-CoV-2'. We apologize for any confusion this may have caused and appreciate your understanding.
In response to your recommendation to add explanations of abbreviations under the tables, we completely agree that this will improve the readability of the manuscript and cater to a broader readership who may not be familiar with these terms. We have included a footnote under each table to clearly define all abbreviations used.
Once again, we appreciate your constructive feedback which will certainly improve the quality and clarity of our manuscript.
Best regards,
Abu Baker Sheikh
Reviewer 2 Report
The aim of the work was to evaluate whether Sars Cov-2 infection had a worse outcome in patients already infected with HIV. It is certainly appreciable to have evaluated a large number of HIV+/Covid versus HIV-/Covid patients. The problem is that the premise is not then reflected in the conclusions, which, by admission of the authors themselves, are not supported by the lack of fundamental data.
Not knowing the immunological status of HIV-infected patients, their viral load, the therapy in progress, does not allow us to outline the viro-immunological picture that would clarify the impact of Covid in this context. In fact, to say that Covid doesn't impact the prognosis of these patients, you need to know whether or not they are immunosuppressed. In this way it can be assessed whether any immunodeficiency worsens the outcome due to less control of the Sars Cov-2 infection or favors it due to a milder inflammatory reaction to Covid.
Unfortunately, the manuscript allows us to have only an epidemiological window, but without scientific repercussions. The same higher incidence in HIV+ of AKI cannot be explained because it is not known whether the patients were treated with TDF disoproxil, a potentially nephrotoxic drug. I also point out that in line 257 of page. 10 risk factors for HIV are listed, including smoking and alcohol, which I do not think are correlated to the virus.
For these reasons, while appreciating the effort of the authors and the large number of patients enrolled, I do not consider the work useful because it does not give exhaustive conclusions.
If the authors recovered the missing data, it could certainly be more valid for publication.
Author Response
Dear Reviewer,
Thank you for your insightful comments and constructive criticism. We appreciate the opportunity to further clarify the aims and impact of our study.
Our study's primary aim was to provide a large-scale epidemiological perspective on the outcomes of HIV-positive patients with SARS-CoV-2 compared to HIV-negative individuals. We acknowledge your valid concerns regarding the lack of individual immunological statuses, viral loads, and ongoing HIV treatments. These parameters indeed provide crucial nuances to understanding the impact of COVID-19 in this context. However, the strength of our study lies in its ability to illuminate population-level trends and outcomes using a substantial data set.
One of the salient findings of our research is the discernible disparities in health outcomes among different demographics. We found a higher proportion of African American individuals and those with lower income statuses among hospitalized HIV-positive COVID-19 patients. This discovery underlines the socio-economic and racial inequalities that impact access to healthcare and disease outcomes—a critical aspect that merits attention and remedial actions.
While stratification based on immunosuppression status could have added further depth to our findings, our study does highlight a critical factor—HIV patients often display elevated complement activation levels, indicative of a persistent hyperinflammatory state, even when their HIV infection is well-managed. This hyperinflammation can have significant implications in the context of a SARS-CoV-2 infection and warrants further investigation, for which our study provides a basis.
[Added in discussion "We acknowledge that our study's database limitations preclude stratification of patients based on CD-4 cell counts. Such an analysis could have nuanced our findings and allowed for an assessment of the effects of COVID-19 on HIV patients with optimally managed conditions. As underscored in research by Rossenheim et al., even well-controlled HIV infections often present an elevated level of complement activation, indicative of persistent hyperinflammation [46]. While antiretroviral therapy (ART) can manage this to some extent, it does not fully resolve the chronic inflammatory state. Such unmitigated inflammation has been linked to a 2-4 fold higher mortality risk from non-AIDS defining events, including cardiovascular disease, and is also associated with metabolic disorders, bone disease, kidney disease, and neurocognitive dysfunction [46].]"
As for the higher incidence of AKI in HIV-positive patients, we concur that knowledge of specific treatment regimens, such as TDF disoproxil usage, would have provided more insights. However, our findings do point towards a significant association worthy of further exploration in studies with access to detailed medication data.
In regard to the mention of smoking and alcohol use as risk factors for HIV on line 257, this was intended to highlight the higher prevalence of these lifestyle factors amongst the younger HIV-positive population, rather than a direct correlation to HIV acquisition. We appreciate your attention to detail and will revise the manuscript to ensure clarity on this point.
[Revised paragraph "Their findings revealed that the average age of HIV patients with COVID-19 was 10 years younger than COVID-19 patients who did not have HIV (55 years vs. 65 years) [23]. This substantial age difference may be influenced by a higher prevalence of behaviors in younger people that increase the risk of HIV transmission. These behaviors can include unsafe sexual practices and sharing of injection equipment, often associated with substance misuse, including smoking, alcohol, and recreational drugs misuse"]
Despite its limitations, our study adds meaningful information to the current literature by revealing key trends and associations at the population level. The findings we have highlighted can guide future, more granular research to further understand the interaction of HIV and COVID-19, particularly in addressing the limitations brought up by your review.
[Revised our conclusion statement "COVID-19 and HIV disease were not associated with higher rates of in-hospital mortality. However, the coexistence of both diseases demonstrates a markedly higher rate of AKI than in the COVID-19 cohort. Our study identified a pronounced racial disparity within the co-infected patient cohort, primarily comprised of African American individuals belonging to lower socioeconomic strata. This striking demographic skew under-scores the intricate interplay between socioeconomic determinants and health outcomes, a finding that is critically relevant amidst the dual challenges posed by HIV and COVID-19. It accentuates the need for a holistic, multi-dimensional approach in the management and mitigation strategies related to these concurrent public health crises, encompassing not only medical but also socio-economic interventions. This approach is paramount in addressing the needs of the most vulnerable populations affected by these diseases. While we acknowledge the limitations of our study in providing a thorough vi-ro-immunological picture due to the lack of specific patient data, such as viral load, CD4 count, and antiretroviral regimen, our findings illuminate key areas requiring focused investigation. Such in-depth research would facilitate the identification of HIV-positive subgroups at an elevated risk for poor clinical outcomes when afflicted with an acute COVID-19 infection. Our findings serve as an impetus for future studies aimed at determining the interplay of these infectious diseases, with a particular emphasis on HIV patients who may be at higher risk for increased in-hospital mortality rates from COVID-19"]
We trust this response satisfactorily addresses your concerns and underscores the potential impact of our research for the broader scientific community. We look forward to receiving any further comments you may have to enhance our manuscript.
Best regards,
Abu Baker Sheikh
Round 2
Reviewer 2 Report
Dear authors, I appreciate the effort to improve the text and I understand that you do not have the viro-immunological data available and that you share my concerns about that. The problem is that without those data, no conclusions can be drawn about the clinical association between Covid-19 and HIV. The number of patients screened is considerable but allows only a summary epidemiological analysis. And as far as the less affluent patients are more affected, that's a bit to be expected. It is also known that the HIV patient basically has a higher level of inflammation and this, in general, may be a factor to consider in the clinical outcome of HIV/Covid-19 coinfection.
In the discussion chapter, on line 325, the authors highlight the result of the study which shows no differences in terms of mortality between the 2 study groups. Before citing other studies in the literature, it should be noted here that these data are only the result of an epidemiological evaluation that does not take into account fundamental data for drawing reliable conclusions on the matter.
I believe that the work can be published, underlining also in the conclusions that there are important bias related to the real understanding of the relationship between the 2 infections and that therefore the study only allows a general epidemiological evaluation of the outcome of subjects with this co-infection which requires further investigations. These will serve to clarify which HIV patient is really most at risk in the event of an overlap of Covid-19 infection.
Author Response
Dear Reviewer,
We appreciate your thoughtful feedback and agree with your point that without detailed viro-immunological data, the conclusions that can be drawn about the clinical association between HIV and COVID-19 are limited. Nonetheless, we believe that our study adds valuable insight into the epidemiological associations between these two diseases and provides a foundation for further investigations.
We also added the suggested lines after citing those mortality studies "It should be noted here that these data are only the result of an epidemiological evaluation"
Our updated conclusion reads as follows:
"While our study did not find a higher rate of in-hospital mortality associated with co-infection of COVID-19 and HIV, it did reveal a markedly higher rate of AKI in the co-infected cohort compared to those with COVID-19 alone. Significantly, our study identified substantial racial and socio-economic disparities within the co-infected patient population, with a majority of patients being African American and belonging to lower socio-economic strata. This finding highlights the critical interplay between socio-economic factors and health outcomes and underscores the need for multi-dimensional strategies that address these concurrent public health crises. These strategies should include not only medical interventions but also socio-economic measures aimed at mitigating the effects of these diseases on the most vulnerable populations. It is crucial to acknowledge that our conclusions are based primarily on an epidemiological evaluation and do not consider certain key viro-immunological factors due to the lack of specific patient data. The unavailability of detailed information such as viral load, CD4 count, and specific antiretroviral therapy regimen limits our ability to draw definitive conclusions regarding the clinical interaction between HIV and COVID-19. Given these limitations, our findings should be viewed as an initial broad epidemiological investigation of the outcomes of individuals co-infected with HIV and COVID-19. Our study suggests areas that warrant more in-depth examination and points towards the necessity for future research to further elucidate the relationship between these two infections. This includes identifying specific HIV-positive subgroups that may be at higher risk of adverse clinical outcomes from COVID-19, thus contributing to a more comprehensive understanding of these intersecting diseases."
We trust that this response adequately addresses your concerns and are open to any further feedback you may have to improve our manuscript.
Best Regards,
Abu Baker Sheikh